# Effect of Material and Process Variables on Characteristics of Nitridation-Induced Self-Formed Aluminum Matrix Composites—Part 2: Effect of Nitrogen Flow Rates and Processing Temperatures

**DOI:** 10.3390/ma13051213

**Published:** 2020-03-08

**Authors:** Dae-Young Kim, Pil-Ryung Cha, Ho-Seok Nam, Hyun-Joo Choi, Kon-Bae Lee

**Affiliations:** School of Advanced Materials Engineering, Kookmin University, Seoul 02707, Korea; kdy6603@kookmin.ac.kr (D.-Y.K.); cprdream@kookmin.ac.kr (P.-R.C.); hsnam@kookmin.ac.kr (H.-S.N.)

**Keywords:** aluminum matrix composites, aluminum nitride, nitridation, exothermic, NISFAC process

## Abstract

The nitridation-induced self-formed aluminum matrix composite (NISFAC) process is based on the nitridation reaction, which can be significantly influenced by the characteristics of the starting materials (e.g., the chemical composition of the aluminum powder and the type, size, and volume fraction of the ceramic reinforcement) and the processing variables (e.g., process temperature and time, and flow rate of nitrogen gas). Since these variables do not independently affect the nitridation behavior, a systematic study is necessary to examine the combined effect of these variables upon nitridation. In this second part of our two-part report, we examine the effect of nitrogen flow rates and processing temperatures upon the degree of nitridation which, in turn, determines the amount of exothermic reaction and the amount of molten Al in the nitridation-induced self-formed aluminum matrix composite (NISFAC) process. When either the nitrogen flow rate or the set temperature was too low, high-quality composites were not obtained because the level of nitridation was insufficient to fill the powder voids with molten Al. Hence, since the filling of the voids in the powder bed by molten Al is essential to the NISFAC process, the conditions should be optimized by manipulating the nitrogen flow rate and processing temperature.

## 1. Introduction

We recently developed the nitridation-induced self-formed aluminum composite (NISFAC) process as a facile and innovative route to manufacturing Al matrix composites (AMCs) [1,2,3]. Whereas, previously, the limited wettability between the reinforcement and the Al matrix had to be overcome by the use of high-energy mechanical stirring (e.g., stir casting) [4,5,6,7,8,9,10], high-pressure infiltration of molten Al into a preform [11,12,13,14,15,16], or high-pressure consolidation of the powder mixture (powder metallurgy) [16,17,18,19,20,21], a key feature of the NISFAC process is its flexibility with respect to the selection of reinforcement regardless of the level of wettability [1,2,3]. Although the nitridation of aluminum has been investigated by a variety of in-situ techniques such as directed metal oxidation [22], pressureless metal infiltration [23], and reactive gas injection [24], etc., there have been limited studies on the nitridation-induced forming of aluminum matrix composites. In particular, we have examined the chemical and structural characteristics of the unique surface morphology of the AMCs produced from the initial powder state in the presence of nitrogen gas and have proposed an explanation for why the AMCs were produced only under a nitrogen atmosphere. In summary, both Al and SiC particles undergo surface modification during heating in a nitrogen atmosphere to improve wettability, resulting in AMCs. This suggests that the properties of the obtained AMCs can be freely tailored by manipulating the type, size, and volume fraction of the reinforcement to generate a number of combinations comparable with those available in alloy design. This also suggests that AMCs with identical properties can be produced with different combinations of Al matrix and reinforcement (or filler). As a result, the number of applicable AMCs can be greatly increased, thereby contributing to the expansion of end-user choice.

Although the nitridation-based NISFAC process is much simpler and easier than conventional processes, the degree of nitridation and, hence, the amount of molten Al generated during the process, is influenced by all relevant variables, including the composition of the Al matrix, the type, size, and volume fraction of reinforcement, and the fabrication temperature and time [1,2,3]. Therefore, it is essential to optimize the process conditions for the selected Al matrix and reinforcement (or filler) system. In the first part of our report, we examined the effects of the size and volume fraction of SiC particles, together with that of the process temperature, upon the Al 6061 alloy matrix composites containing SiC [3]. There we found that the most important factor for producing high-quality AMCs is the generation of sufficient molten Al to fill the voids in the powder bed. The amount of molten Al formed during heating in a nitrogen atmosphere is also determined by the temperature inside the powder bed, which is governed by the degree of nitridation and the process temperature. As mentioned earlier, nitridation is affected by almost all manufacturing parameters. Therefore, the effects of nitrogen flow rate and process temperature upon the production of AMCs consisting of the same Al 6061 alloy matrix with an initial average Al powder size of ~7 μm and containing SiC with an initial average powder size of ~40 μm are examined in the present work.

## 2. Materials and Methods

The composites were produced in the same manner as in the first part of this study; Al 6061 alloy was selected as the matrix and SiC particles were selected as the reinforcement. The average particle size of the Al 6061 alloy powder was ~7.18 μm (Chengdu Best New Materials Co., Ltd, Sichuan, China). The average size of the SiC particles was 40 μm (Showa Denko, Toyama, Japan) and the volume fraction was fixed at 20%. The Al 6061 alloy powder and SiC particles were first mixed using a Turbula mixer (DM-T2, Daemyoung enterprise Co., LTD., Gwangmyeng, Korea). The mixture was then placed in a graphite crucible, charged in a furnace, heated in a nitrogen atmosphere at 610 to 650 °C for 20–60 mins, then removed from the furnace and air cooled. The heating rate was 5 °C/min, the nitrogen flow rate was 1–4 L/min, and the exhaust was conducted via a beaker filled with water to suppress the ingress of external oxygen. In addition, since the production of AMCs was significantly affected by the reaction between the Al powder and nitrogen gas (nitridation) during heating in a nitrogen atmosphere, the crucible weight was measured before and after heating in order to calculate degree of nitridation. The temperature change inside the powder bed due to the exothermic nitridation reaction was also measured with the aid of data acquisition software (Lutron, SW-U801-WIN) by inserting a thermocouple into the center of the powder bed. The phase-change behavior according to process temperature and nitrogen concentration was analyzed using an X-ray diffractometer (XRD, Rigaku D-max 2500, Tokyo, Japan). The microstructure of the prepared AMCs was observed using an optical microscope (Eclipse LV100ND, Nikon, Japan) and a scanning electron microscope (SEM, JSM 2001F, JEOL, Akishima, Japan) with energy-dispersive X-ray spectroscopy (EDS).

## 3. Results and Discussion

The key to determining the success of the NISFAC process is the formation of sufficient molten Al to fill the voids inside the powder bed. In particular, when the production temperature is lower than the melting point (or liquidus) of Al (or alloy), the Al powder can only melt when the temperature is locally increased due to the exothermic nitridation reaction. Hence, the amount of molten Al is determined by the degree of nitridation, which can be controlled by the various process parameters.

### 3.1. Effect of Nitrogen and Argon Gas

Figure 1a shows the internal temperature of the powder bed when heating a mixture of Al 6061 alloy and SiC (40 μm, 20 vol.%) from room temperature (RT) to 640 °C for 1 h under an atmosphere of argon or nitrogen (2 L/min). For both types of atmosphere, the temperature of the powder bed increased at approximately the same rate during the heating from RT to 610 °C. However, when the temperature of the powder bed exceeded 610 °C, a more rapid temperature increase was observed under the nitrogen atmosphere than the argon atmosphere. This is due to the exothermic nature of the nitridation reaction. Furthermore, under the nitrogen atmosphere, the temperature rise of the powder bed became rapid after reaching approximately 645 °C, and subsequently became more gradual as the temperature approached the maximum value of about 680 °C.

Photographic images of the air-cooled crucibles after heating for 1 h under a nitrogen or argon atmosphere, respectively, are presented in Figure 1b,d, while images of the respective samples after machine working are presented in Figure 1c. For the sample heated in nitrogen, the significant amount of shrinkage observed after cooling (Figure 1b) suggests that sufficient molten Al was produced during heating. Moreover, a bright metallic luster was obtained for this sample after lathe working (Figure 1c), demonstrating the production of a sound composite. When heated in argon, the maximum temperature inside the bed was 631 °C, which is lower than the programmed furnace target temperature of 640 °C but higher than the solidus of the Al 6061 alloy (582 °C), hence melting still occurred. However, unlike the powder heated in nitrogen, little shrinkage was observed after solidification (Figure 1d) and molten Al alloy leaked out of the bed surface (Figure 1e) due to the poor wettability between the molten Al and SiC [2,3] as in the previous studies. Hence, high-quality composites were not produced in this case.

Figure 2 indicates the effect of varying the amount of nitrogen during the production of the same composites by heating the mixtures for 1 h at a set temperature of 640 °C. The change in temperature inside the powder bed is seen to vary according to the nitrogen flow rate (1 to 4 L/min) and, hence, the amount of nitrogen supplied. At nitrogen flow rates of 1–3 L/min, the temperature increase was first gradual and became rapid at around 645 °C due to the exothermic nitridation reaction. Furthermore, as the amount of nitrogen increased, the rapid temperature rise occurred sooner, but the maximum temperature was significantly decreased. We reported in the previous study that the temperature at the center of the bed was very high compared to that at the bottom or top during the nitridation of Al powders [25]. This indicates that nitridation begins at the center of the powder bed. Nitrogen gas supplied into the chamber passes through the surface of the powder bed and moves inwards through the pore network. Nitrogen reaching the top of the bed is soon discharged through the exhaust pipe. Hence, contact between nitrogen and the Al particle surface is maintained by the continuous supply of fresh nitrogen gas into the chamber. Therefore, nitridation does not occur at the top of the powder bed because the contact time between the nitrogen gas and the Al powder is very short there. By contrast, since the nitrogen gas reaching the center of the bed via the pore network remains in a relatively closed area, nitridation can be initiated during a longer contact time relative to the upper and lower portions of the bed for a given flow rate. Thus, increasing the amount of nitrogen allows a high nitrogen concentration to be maintained in a relatively wider volume in the center of the bed (Figure 2b). Hence, as the quantity of Al particles experiencing nitridation increases, the heat dissipated by the exotherm increases, and the temperature of 645 °C is reached sooner, at which stage a rapid temperature rise occurs. This can be seen from the fact that the bed temperature before the onset of rapid temperature rise is relatively higher as the amount of nitrogen increases, as shown in Figure 2a. As will be explained later, however, no rapid temperature rise was observed when the nitrogen flow rate was increased to 4 L/min.

The degrees of nitridation and the maximum temperatures of the powder bed under various nitrogen flow rates are indicated in Appendix A along with photographic images of the composites in their crucibles and after lathe working. As the inlet flow rate and, hence, the amount of nitrogen supplied increases, the degree of nitridation and maximum temperature both decrease. Since the amount of AlN formed by the nitridation reaction affects the properties of the composite, the amount of AlN should be controlled according to the requirement of the end user.

Figure 3 presents optical micrographs of the composites prepared under different nitrogen flow rates. Uniform distribution of SiC is observed for all samples and no significant microstructural differences were detected. In addition, the XRD patterns in Figure 4 indicate that AlN was formed in all the composites regardless of nitrogen flow rate and that other reaction products such as Al_4_C_3_ were not detected. In particular, Al_4_C_3_ was not detected at a nitrogen flow rate of 1 L/min, even though the maximum temperature was approximately that at which this carbide can form (742 °C). As indicated in the above discussion, the temperature rise was localized and the holding time for this temperature range was evidently too short for carbides to form. In this study, composites were prepared at 640 °C, which is below the liquidus (652 °C) of the Al 6061 alloy. Hence, it is possible to produce the composite even at a temperature below the liquidus line, which is one of the advantages of the NISFAC process. Unlike conventional processes involving liquid phases, composites can be produced even below the melting point (or liquidus) of Al and its alloys, which not only reduces the formation of undesirable reactants (e.g. Al_4_C_3_ when SiC is used), but also reduces the energy-consumption during the manufacturing processes.

To examine in detail the effect of nitrogen concentration upon the composite production, the change in the microstructure of the powder bed was tracked during heating under nitrogen flow rates of 1 and 4 L/min at a set temperature of 640 °C. The degrees of nitridation and powder bed temperatures of the composites produced with various nitrogen flow rates and heating times are presented in Table 1, while Appendix A presents photographic images of the respective powders in their crucibles (rows I and V) and in the powder beds (rows II and VI), along with images of the samples after cutting (rows III and VII) and after lathe working (rows IV and VIII). In addition, SEM images of the powder bed microstructures observed at various heating times are presented in Figure 5. It can be seen that a nitrogen flow rate of 1 L/min resulted in a very small degree of nitridation (0.1%) after heating for 15 mins with a set temperature of 640 °C, so that the sample remained in the powder form. This is because the temperature inside the powder bed remained relatively low (~ 622 °C) and, although this was above the Al 6061 alloy solidus (582 °C), almost no molten Al was formed (Figure 5a and Appendix A, row II).

After 32 minutes of heating, the degree of nitridation increased to 2.9% and the powder bed temperature increased to 639 °C. However, the amount of molten Al phase was not enough to form highly dense composites and the final composites were easily broken after solidification (Appendix A, row II). In addition, the arrows in Figure 5b,c indicate that molten Al was released from the powder and covered the surfaces of adjacent Al and SiC particles. Molten Al was thus in intimate contact with a very large surface area of the SiC particles. The melting of the Al powder was caused by the localized increase in temperature above the Al melting temperature due to the exothermic nitridation reaction.

After heating for 44 mins, the top of the powder bed remained in powder form (Appendix A, row I), whereas sufficient shrinkage occurred at the bottom of the powder bed to allow complete solidification (Appendix A, row II). This was because the powder bed temperature reached ~ 649 °C or more, which is close to the liquidus temperature of the Al 6061 alloy (652 °C), so enough molten Al was formed to fill the voids in the powder bed (Figure 5e,f). However, as can be seen from the cross-section image in Appendix A (row III) and the SEM image in Figure 5d, some un-melted areas were present near the surface of the powder bed.

With a nitrogen flow rate of 1 L/min, the temperature inside the bed continued to rise after 60 mins of heating (Figure 2), hence the nitridation reaction continued. This can also be seen in the cross-section image in Appendix A (row III) and the SEM images in Figure 5g,h. This indicates that much more melt was formed in the center than at the top of the bed. In addition, the amount of molten Al was significantly increased relative to that observed after 50 mins of heating (Figure 5f,h). As shown in the crucible image, some powder remained on the top even after heating for more than 1 h, but the bed temperature decreased after reaching the maximum temperature of 742 °C while the amount of molten Al increased (Figure 5i).

The temperature of the bed and the degree of nitridation for the composites after heating for 24 mins under a nitrogen flow rate of 4 L/min were ~ 639 °C and 2.9%, respectively. These values were similar to those obtained after heating for 32 mins under a flow rate of 1 L/min. In addition, the crucible images indicate a larger shrinkage after 24 mins of heating at 4 L/min than after 32 mins of heating at 1 L/min, and that the amount of powder remaining on the bed was also reduced and completely solidified at the higher flow rate (Appendix A, row V). The images obtained after cutting show similar morphologies after 44 mins of heating at both flow rates. In addition, after 44 mins of heating at 4 L/min, the peak temperature of 641 °C has already been passed so that sufficient shrinkage has occurred and the amount of powder remaining on the upper part of the bed has been greatly reduced (Appendix A, rows V-VIII).

As shown in Figure 6, these macroscopic changes were also reflected in the microstructure. After heating for 44 mins, the amount of molten Al obtained at 4 L/min was greatly increased relative to that obtained at 1 L/min. Since the rate of nitridation at a nitrogen flow rate of 4L/min is greater than that at 1 L/min, the temperature inside the powder bed increases more rapidly and molten Al is formed relatively faster. As a result, the molten Al fills the pore space inside the bed and blocks the nitrogen supply, suppressing further nitridation and preventing a rapid temperature rise. By contrast, at a nitrogen flow rate of 1 L/min, the nitridation rate is slow, hence the formation of the liquid phase is slow, nitrogen is supplied into the bed for a relatively longer time, the nitridation time increases and, hence, the temperature rises rapidly due to the increase in the degree of nitridation.

### 3.2. Effect of Heating Temperature at a Fixed Nitrogen Flow Rate

Since the rapid temperature rise was not observed when heating at a nitrogen flow rate of 4 L/min, the composite was prepared at a fixed flow rate of 4 L/min while heating for 1 h at set temperatures ranging from 610 to 650 °C in order to analyze the effect of the production temperature. Figure 7 shows the temperature changes measured inside the powder bed for various heating temperatures. For all production temperatures, each graph displays several changes of slope such that the second change of slope (marked by the arrows in Figure 7) indicates the temperature rise due to nitridation. The temperature rise stems from two main reasons, namely: i) the exothermic nitridation reaction and ii) the rise in temperature the furnace towards the set temperature. Therefore, the temperature rise due to nitridation may be most clearly distinguished at the lowest set temperature of 610 °C. As shown in Figure 7, at 610 °C, the temperature gradient changes near the solidus temperature (582 °C) of the Al 6061 alloy, indicating that the nitridation of Al begins at very low temperatures. Thus, the time for initiation of nitridation (as indicated by the second change in slope) decreases with increasing production temperature, but the powder bed temperature then increases due to the relatively increased effect of heating. In contrast to the temperature behavior when heating to set temperatures in the range of 610 to 640 °C, heating to 650 °C generated a sharp internal temperature change at about 645 °C, as was observed in Figure 2. The peak internal temperatures together with the set temperatures are summarized in Appendix A along with photographic images of the composites in the crucibles and after machine working. Thus, the peak internal temperature did not vary significantly with set temperatures except for the set temperature of 650 °C, which exhibited the sharp increase in powder bed temperature. The composites prepared at a set temperature of 610 °C (Appendix A, row II) exhibited a reduced metallic luster and a smaller amount of shrinkage compared to those produced at higher set temperatures. Therefore, in order to manufacture a high-quality composite, manufacturing temperatures of 620 °C or higher are preferable. In addition, since the maximum temperature inside the powder bed remains below the liquidus of the Al 6061 alloy (652 °C) at set temperatures in the range of 620 to 640 °C, this can greatly limit the formation of undesirable reaction products. When the heating time was increased to 90 mins for the set temperature of 610 °C, however, the crucible shrinkage and metal gloss increased significantly (Appendix A, row I). As described above, when manufacturing the composite using the NISFAC process, it is possible to control the amount of molten Al by adjusting the process variables, so that the composite may be manufactured under a variety of conditions. This diversity of processing is another advantage of the NISFAC process. That is, even for the same composite system (e.g., the Al/SiC system studied herein), it is possible to tailor the characteristics of the final product by controlling various process parameters such as the composition of the selected Al alloy, the matrix powder size, the SiC particle size, the volume fraction of SiC, the manufacturing time and temperature, the nitrogen concentration, etc.

## 4. Conclusions

We have reported the effects of process parameters upon the fabrication of Al 6061/SiC composites using the NISFAC process. In this second part of a two-part report, the effects of nitrogen flow rate and heating temperature upon the nitridation behavior of the composites were investigated. When the nitrogen flow rate was too slow, or the set temperature was too low, nitridation did not occur sufficiently to form enough molten Al to fill the voids in the powder and high-quality composites were not obtained. In summary, the production temperature and time, the size and volume fraction of the SiC particles, and the amount of nitrogen all influence the production of high-quality composites. After years of developing the NISFAC process, we have manufactured composites with various reinforcement materials and various Al matrices. The greatest advantage of the Al nitridation-based NISFAC process is that it can produce sound composites regardless of the wettability between the Al matrix and the reinforcement. This means that it is possible to produce countless combinations of composites that have only been theoretically possible so far. Over the years, research has shown that the most important part of the NISFAC process is the formation of sufficient liquid Al to fill the voids inside the powder bed. In addition, since the amount of liquid Al is affected by the various process parameters affecting the degree of nitridation, a systematic study of each system is essential. Such research will allow us to freely manufacture composites with properties suitable for the end use (as in the design of alloys) by establishing the optimum process conditions.

## Figures and Tables

**Figure 1 materials-13-01213-f001:**
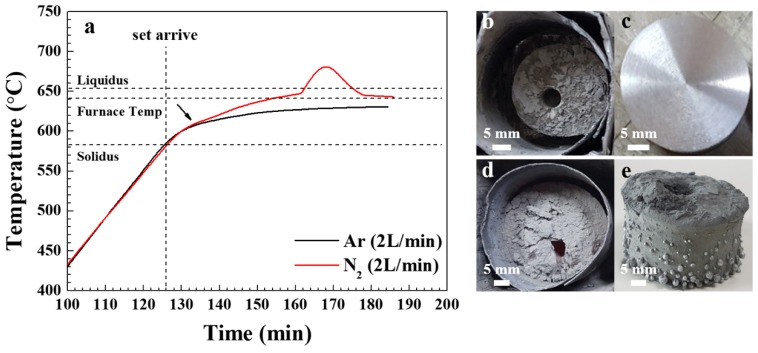
(**a**) The temperature variation of 20 vol.% SiC (40 μm)/Al 6061 composites during heating at a set temperature of 640 °C under nitrogen and argon atmospheres (flow rate: 2 L/min); (**b–e**) Photographic images of the composites produced under an atmosphere of nitrogen (**b**,**c**) and argon (**d**,**e**) in the air-cooled crucibles (**b**,**d**) and after lathe working (**c**).

**Figure 2 materials-13-01213-f002:**
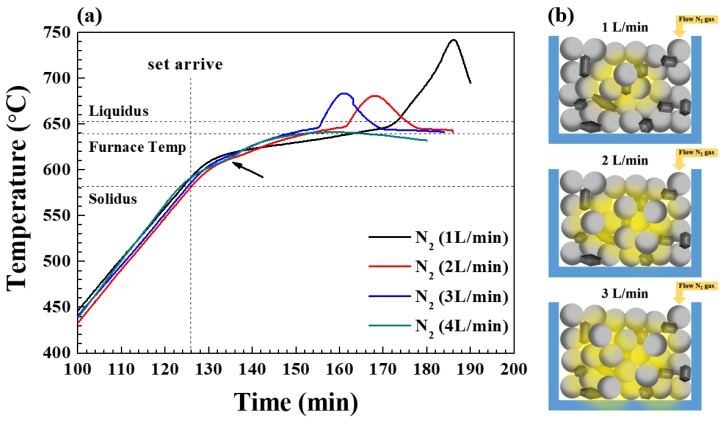
(**a**) The temperature variation of 20 vol.% SiC (40 μm)/Al 6061 composites during heating for 1 h at a set temperature of 640 °C under a nitrogen atmosphere at flow rates of 1-4 L/min. (**b**) A schematic drawing of the relative concentration of nitrogen in the center of powder bed with increasing flow rates of nitrogen gas.

**Figure 3 materials-13-01213-f003:**
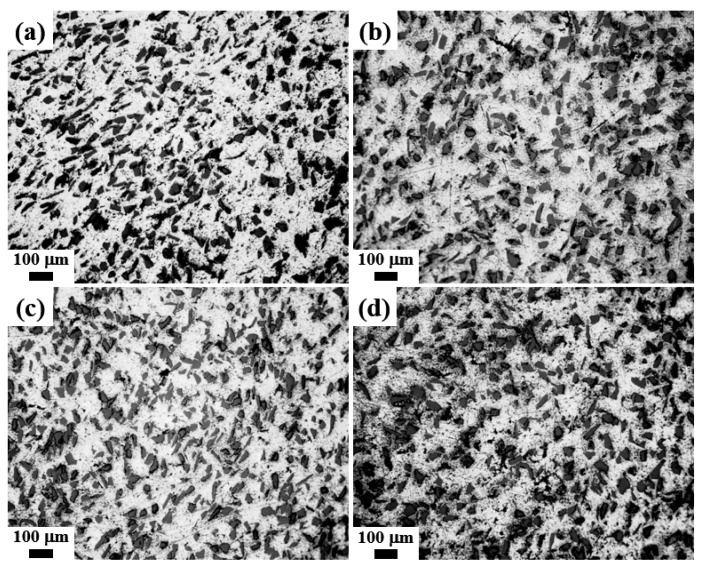
Optical micrographs of 20 vol% SiC (40 μm)/Al 6061 composites produced by heating for 1 h at 640 °C under various flow rates of nitrogen gas: (**a**) 1 L/min, (**b**) 2 L/min, (**c**) 3 L/min, and (**d**) 4 L/min.

**Figure 4 materials-13-01213-f004:**
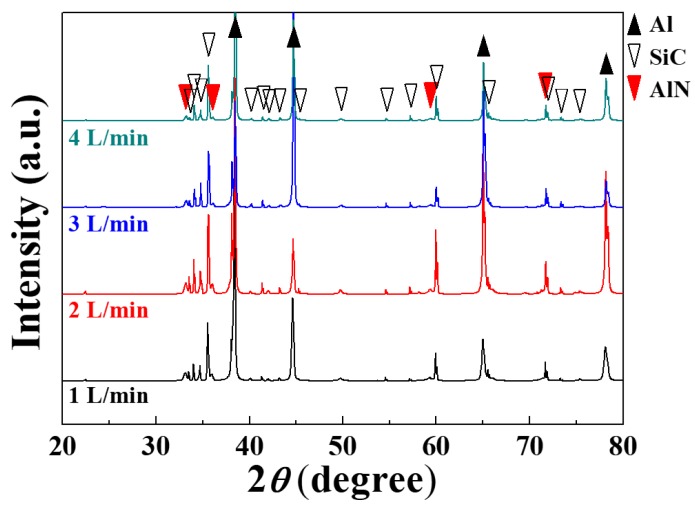
XRD patterns of 20 vol% SiC (40 μm)/Al 6061 composites produced by heating for 1 h at 640 °C under various flow rates of nitrogen gas.

**Figure 5 materials-13-01213-f005:**
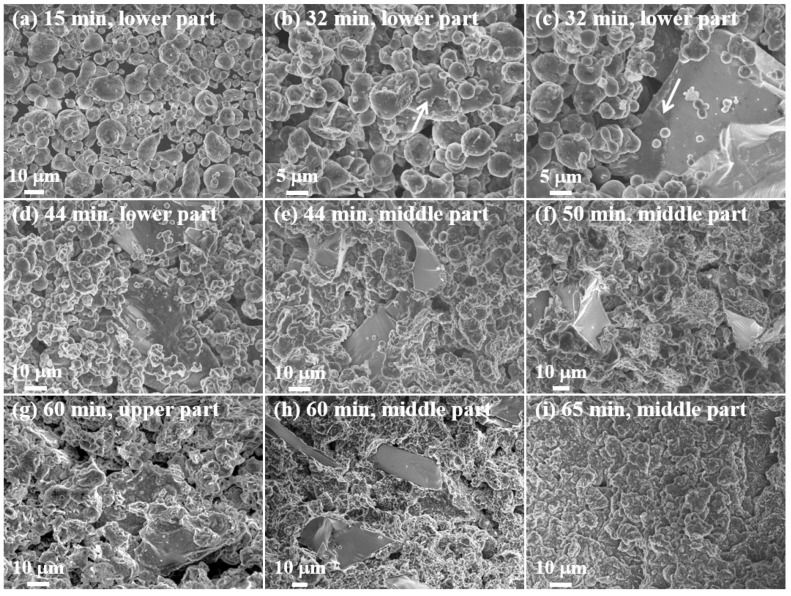
SEM images of the 20 vol.% SiC (40 μm)/Al 6061 composites at various positions in the powder bed after various times of heating at a set temperature of 640 °C under a nitrogen flow rate of 1 L/min: (**a**–**d**) lower part, (**e**,**f**,**h**,**i**) middle part, and (**g**) upper part.

**Figure 6 materials-13-01213-f006:**
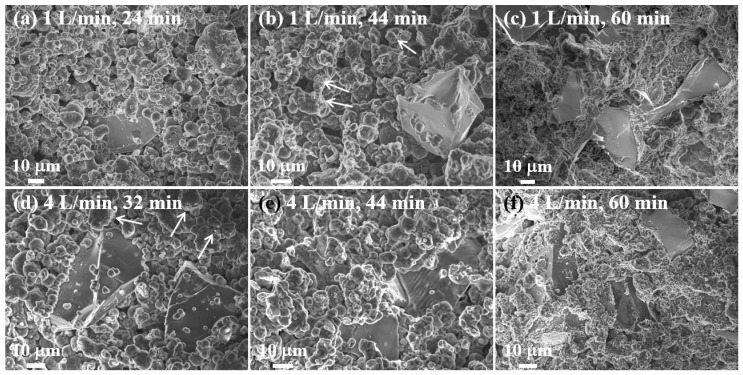
SEM images of 20 vol.% SiC (40 μm)/Al 6061 composites in the powder bed after heating for various times at a set temperature of 640 °C under nitrogen flow rates of 1 L/min (**a**–**c**) and 4 L/min (**d–f**).

**Figure 7 materials-13-01213-f007:**
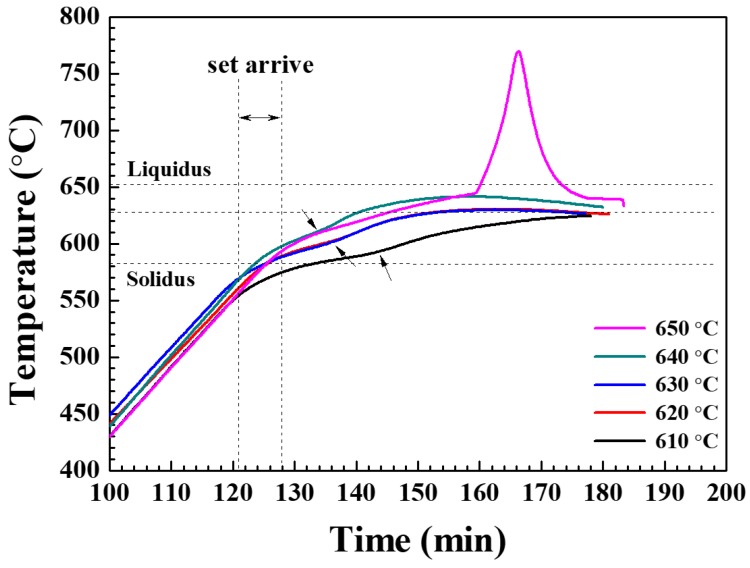
The temperature variation of 20 vol.% SiC (40 μm)/Al 6061 composites during heating at various set temperatures (610–650 °C) under a fixed nitrogen flow rate of 4 L/min.

**Table 1 materials-13-01213-t001:** The degrees of nitridation and temperatures of the powder bed with various nitrogen flow rates and heating times.

Flow Rate of N_2_ Gas (L/min)		Heating Time (min)						
		15	24	32	44	50	56	60
**1**	Bed temperature (°C)	622		639	649	654	654	
	Degree of nitridation (%)	0.1		2.9	4.8	5.0		8.7
4	Bed temperature (°C)		639		639			641
	Degree of nitridation (%)		2.9		6.3			8.1

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
