# Peer review of "Effect of Material and Process Variables on Characteristics of Nitridation-Induced Self-Formed Aluminum Matrix Composites—Part 2: Effect of Nitrogen Flow Rates and Processing Temperatures"

_materials, 2020, doi:10.3390/ma13051213_

Round 1

Reviewer 1 Report

Considering the materials-702592 manuscript together, the following remarks can be made:
1. In the abstract, the introduction is again duplicated, including the justification of relevance. Moreover, the relevance has not been substantiated.
2. Figure 7 - are these exactly composites? According to SEM images, the material is similar to powder. At the same time, the density is not mentioned in both works. Density will significantly affect the final properties of the composite.
3. Again, many pictures of the samples. For what? This is a scientific article, not a scientific and technical report on the work done.
4. There are no scientific conclusions in the work, only technical ones.
5. In both studies, a deeper study of the structure of composites at the aluminum – silicon carbide – aluminum nitride phase boundaries is lacking.
In my opinion, the presented results are not enough for two articles. With a sufficient analysis of the results, one could write one, but of a better quality, scientific article.

Reviewer 2 Report

This article is second part of the report focusing on the nitrogen flow and temperature regimes on metal matrix formation. Article presents extensive experimental results, but in my opinion lacks general conclusions of the feasibility of the presented method.

Experiments with different N flowrates are done. Schematics of the experimental setup would be helpful showing the location of the nitrogen inflow and outflow, container size, heater ect. Fig 2(b) Nitrogen content is proportional to the red doted circle ? Not a good representation. 3,Fig.6, Fig.10. It is pointless to show many similar pictures with similar metal cylinders. Better include these results in graph or table. 4. It is only said that microstructure is similar. In conclusions authors says that if N flow or temperature is too low, it leads to lower material quality. How material quality is defined and what is the aim ? Fig 7 shows material structure at different places and different processing times. It would be better to use EDX imaging to distinguish Si and Al phases. Currently the results from these SEM images are not sufficiently explained.

Reviewer 3 Report

The research work presented in the manuscript is dealing with the obtaining of AlMMC through a novel in-situ method that was previously disclosed in a recent US patent and paper by the same authors. Considering the simplicity of the method with such good results the subject of the manuscript is of high importance both to science and industry. Present paper is discussing the influence of the nitrogen flow rate and process temperature on the quality of the obtained composite material. The presentation of the experimental method and results is sufficient in regards to the goal of the authors. However, there are some aspects that can be improved in present or future presentation of the different aspects of the method:

The content of past results is not sufficiently mentioned as there are different processes already applied industrially that are similar to the one presented here. The authors should have also mentioned important in-situ methods that deal with gas/melt interactions to form ALMMC, such as: DIMOX, PRIMEX, Osprey, Scholz - Hou method or the more recent reactive gas injection (RGI) method. The introduction is expected to contain these references too. While the paper is discussing two important process parameters to determine the optimum results a more scientific aspect would be represented by the phenomenons that take place at particle interfaces during nitridation process, which are key to the whole process. In this respect reaction thermodynamic and kinetics studies are necessary.  While the obtaining of a great quality composite material through the presented method is very surprising, the paper is not discussing at all the porosity found in the material, knowing that a fair volume of gases is used during the process. SEM images show a sinterlike structure with porosity present in the material. In order to determine the viability of the method mechanical tests are required for the resulted composite material Half of the conclusions section is not pertinent to the present research results but to the overall method presentation and needs major improvement.

Round 2

Reviewer 1 Report

I thank the authors for a detailed explanation.